



**Measurement report: Diurnal variations of brown carbon during two distinct seasons in a megacity in Northeast China**

Yuan Cheng[1], Xu-bing Cao[1], Jiu-meng Liu[1,*], Ying-jie Zhong[1], Qin-qin Yu[1], Qiang Zhang[2] and Ke-bin He[3]

[1] State Key Laboratory of Urban Water Resource and Environment, School of Environment, Harbin Institute of Technology, Harbin 150090, China

[2] Ministry of Education Key Laboratory for Earth System Modeling, Department of Earth System Science, Tsinghua University, Beijing 100084, China

[3] State Key Joint Laboratory of Environment Simulation and Pollution Control, School of Environment, Tsinghua University, Beijing 100084, China

[*]Corresponding author. Jiu-meng Liu (jiumengliu@hit.edu.cn).

**Abstract**

Brown carbon (BrC) represents an important target for the "win-win" strategy of mitigating climate change and improving air quality. However, estimating co-benefits of BrC control remains difficult for China, partially because current measurement results are insufficient to represent the highly variable emission sources and meteorological conditions across different regions. In this study, we investigated, for the first time, the diurnal variations of BrC during two distinct seasons in a largely unexplored megacity in Northeast China. The winter campaign conducted in January of 2021 was characterized by low temperatures rarely seen in other Chinese megacities (down to about –20 °C). The mass absorption efficiencies of BrC at 365 nm ($MAE_{365}$) were found to be ~10% higher at night. The variations of $MAE_{365}$ could not be explained by the influence of residential biomass burning emissions or secondary aerosol formation, but were strongly associated with the changes of a diagnostic ratio for the relative importance of coal combustion and vehicle emissions ($R_{S/N}$). Given that most coal combustion activities were uninterruptible, the higher nighttime $MAE_{365}$ in winter were attributed primarily to increased emissions from heavy-duty diesel trucks. The spring





campaign conducted in April of 2021 was characterized by frequent occurrences of agricultural fires,
as supported by the intensive fire hotspots detected around Harbin and the more-than-doubled
levoglucosan to organic carbon ratios (LG/OC) compared to winter campaign. In spring, $MAE_{365}$
depended little on $R_{S/N}$ but exhibited a strong positive correlation with LG/OC, suggesting open
burning emissions as the dominant influencing factor for BrC's light absorption capacity. $MAE_{365}$
were ~70% higher at night for the spring campaign, pointing to the prevalence of nighttime
agricultural fires, which were presumably in response to local bans on open burning. It is noteworthy
that the agricultural fire emissions resulted in distinct peak at ~365 nm for the light absorption
spectra of BrC, and a candidate for the compounds at play was inferred to be $C_7H_7NO_4$. Due to the
presence of the ~365 nm peak, the absorption Ångström exponents could not be properly determined
for the agricultural fire-impacted samples. In addition, the ~365 nm peak became much less
significant during the day, likely due to photo-bleaching of the relevant chromophores.



## 1. Introduction

Light-absorbing organic carbon, i.e., brown carbon (BrC), exerts important yet poorly understood effects on climate and the environment (Brown et al., 2018; Zeng et al., 2020; Sand et al., 2021). As a mixture of numerous organic compounds from both primary emissions and secondary formation, BrC exhibits extreme complexity in spectroscopy, composition and evolution (Laskin et al., 2015; Brege et al., 2021; Washenfelder et al., 2022). Measurement techniques for BrC absorption mainly fell into two categories, including solvent extraction followed by light absorption spectrum measurement (Chen and Bond, 2010; Hecobian et al., 2010) and apportionment of total aerosol absorption to the contributions from black carbon and BrC (Yang et al., 2009; Lack et al., 2012). So far, consistency between BrC results from these two types of approaches has not been addressed, with variable relationships, either linear or non-linear, and unclear influencing factors (Kumar et al., 2018; Zeng et al., 2022). This inconsistency introduced substantial difficulties to the integration of BrC measurement results across studies and regions (Wang et al., 2022), which is essential for unfolding the links between BrC sources and optical properties. In addition, efforts were also made to explain BrC absorption on a molecular level. Several techniques were shown to be powerful, such as electrospray ionization Fourier transform ion cyclotron resonance mass spectrometry (ESI FT-ICR MS; Wozniak et al., 2008; Jiang et al., 2021; Zeng et al., 2021), high performance liquid chromatography coupled with high resolution mass spectrometry (HPLC/HRMS; Lin et al., 2018; Huang et al., 2022; Xu et al., 2022), and two-dimensional gas chromatography with time of flight mass spectrometer (GC×GC-ToF-MS; Huo et al., 2021). These techniques were more frequently applied to laboratory-generated primary or secondary BrC (e.g., Lin et al., 2015), which usually had less complex composition than ambient BrC and thus showed



relatively high fraction of resolvable chromophores, e.g., up to ~85% for those emitted by biomass
burning (Huang et al., 2022).

The absorbing nature of BrC makes it a non-negligible contributor to positive radiative forcing

(Saleh, 2020), while the considerable contribution of organic aerosol to fine particulate matter
($PM_{2.5}$) makes BrC an important source of air pollution (Wang et al., 2019). Consequently, BrC
represents a key species for the "win-win" strategy of mitigating climate change and improving air
quality. Given the highly variable emission sources and meteorological conditions across different
regions in China, field observational results on BrC are far from being enough to constrain air
quality and climate models, limiting the ability to evaluate the co-benefits of BrC control. In this
study, we focused on a largely unexplored city cluster, the Harbin-Changchun (HC) metropolitan
area in Northeast China. Compared to other regions with intensive studies of BrC as well as other
air pollutants (e.g., the North China Plain), HC was characterized by extremely cold winter and
strong impacts of biomass burning on top of other anthropogenic emissions (e.g., from coal
combustion). The first feature was related to the relatively high latitudes of HC. For example, as the
northernmost megacity in China, Harbin has an average temperature of about –20°C in January,
significantly lower than that of Beijing. The second feature was related to the massive agricultural
sector in HC. Until recently, open burning was still an irreplaceable approach for the disposal of
crop residues in this region, presumably because the amount of agricultural wastes were too huge
for the capacity of sustainable use. The agricultural fires frequently resulted in heavily-polluted
episodes with high $PM_{2.5}$ concentrations rarely encountered in other Chinese megacities (e.g.,
hourly-average of ~1000 μg/m$^3$ in Harbin; Li et al., 2019). These two features highlighted the
uniqueness of HC for haze studies in China.





This measurement report, for the first time, presented field observational results on the diurnal
variations of BrC during two distinct seasons, i.e., a frigid winter and an agricultural fire-impacted
spring, in the central city of HC. Drivers for the diurnal variations were discussed based on
indicators of various sources. Particularly, the agricultural fires were found to result in unique
absorption spectra of brown carbon. This study provided implications for parameterization of BrC
in climate models.
**2. Methods**
**2.1 Field sampling**
Daytime and nighttime $PM_{2.5}$ samples were collected on the campus of Harbin Institute of
Technology (HIT) during winter and spring of 2021. HIT was surrounded by residential and
commercial areas, without major industrial sources nearby, and thus represented a typical urban site.
The sampling was done by a mass flow controlled high-volume sampler (TE-6070BLX-2.5-HVS;
Tisch Environmental, Inc., OH, USA), which was operated at a flow rate of 1.13 $m^3$/min using pre-
baked quartz-fiber filters (8″ × 10″, 2500 QAT-UP; Pall Corporation, NY, USA). Daytime and
nighttime samples were collected from 9:00 to 16:00 and from 21:00 to 5:00 of the next day,
respectively. The winter campaign covered the entire January of 2021, and the spring campaign was
conducted during 10–30 April, 2021.
**2.2 Laboratory analysis**
Two punches with diameters of 20 mm were taken from each sample, combined and then
extracted by deionized water. The water extract was analyzed using a Dionex ion chromatography
system (ICS-5000⁺; Thermo Fisher Scientific Inc., MA, USA). Levoglucosan, an organic tracer for
biomass burning, was determined by the high-performance anion-exchange chromatography





coupled to pulsed amperometric detection (HPAEC-PAD) method (Engling et al., 2006; Yttri et al.,
2015). Inorganic ions such as nitrate, sulfate, chloride, ammonium and potassium were also
measured. Linear regression of the total cation concentration on that of total anion (both in $\mu eq/m^3$)
led to a slope of 1.14 ±0.01 (intercept was set as zero; $r = 0.99$), indicating a neutralized feature of
the Harbin aerosols.

Two punches with diameters of 47 mm were taken from each sample and used to determine

carbon fractions. One punch was directly measured for organic carbon and elemental carbon, while
the other punch was immersed in methanol (HPLC grade; Fisher Scientific Company L.L.C., NJ,
USA) for an hour without stirring or sonication, dried in air for another hour, and then analyzed.
Both punches were measured by a Thermal/Optical Carbon Analyzer (DRI-2001; Atmoslytic Inc.,
CA, USA), which was operated with two commonly-used temperature protocols (i.e., IMPROVE-
A and NIOSH) and transmittance charring correction. The difference of total carbon (TC)
concentrations between the untreated and extracted punches ($TC_{untreated} - TC_{extracted}$) was used to
represent the amount of organic carbon that is soluble in methanol (MSOC), following the method
developed by Chen and Bond (2010) and refined by Cheng et al. (2016). Given that the TC
measurement was independent of the temperature protocol used, both $TC_{untreated}$ and $TC_{extracted}$ were
determined as the averages of total carbon results from IMPROVE-A and NIOSH. A benefit of this
approach was that the uncertainty of MSOC ($\sigma$) could be estimated for each sample based on the
parallel TC measurements by different protocols:
$\sigma = \sqrt{\left(SD\ of\ TC_{untreated}\right)^2 + \left(SD\ of\ TC_{extracted}\right)^2} \Big/ \left(TC_{untreated} - TC_{extracted}\right)$
where SD indicates standard deviation. In this study, $\sigma$ averaged 3.3 ±2.9% with a median of 2.4%.
In addition, organic compounds that are in-soluble in methanol, i.e., MIOC, was measured as the



organic carbon concentration of the extracted punch. Unless stated otherwise, (i) OC involved in
the following discussions indicates the sum of MSOC and MIOC, and correspondingly, EC indicates
elemental carbon measured by the extracted punch; and (ii) all the carbonaceous aerosol
concentrations are based on IMPROVE-A, except MSOC which did not rely on analytical protocol.
The MSOC to OC ratios averaged 0.90 ± 0.05, indicating an overall high extraction efficiency of
methanol for dissolving organic aerosols.

Light absorption spectra of the methanol extracts were measured over the wavelength ($\lambda$) range

of 200–1110 nm, using a spectrophotometer coupled with a 2.5-m long liquid waveguide capillary
cell (LWCC; World Precision Instrument, FL, USA). The spectrophotometer, consisting of a DH-
mini UV-VIS-NIR light source and a Maya2000 Pro spectrometer (Ocean Optics Inc., FL, USA),
provided wavelength-resolved optical attenuation ($ATN_\lambda$) of the dissolved BrC, which could then
be converted to BrC absorption coefficient [$(b_{abs})_\lambda$] (Hecobian et al., 2010). The ratio of $(b_{abs})_\lambda$ to
MSOC concentration was considered the bulk mass absorption efficiency ($MAE_\lambda$) of brown carbon,
given the close-to-one MSOC/OC. The wavelength dependence of BrC absorption was determined
based on $\ln(ATN_\lambda)$ and $\ln(\lambda)$, and was expressed as the absorption Ångström exponent (AAE). The
AAE calculation was performed over 310–460 nm, the same $\lambda$ range adopted by previous studies
conducted at the same site using the same laboratory analysis procedures (Cheng et al., 2022a).
**2.3 Additional data sets used**

Air quality data and meteorological data were obtained with a time resolution of 1 hour from

the China National Environmental Monitoring Center (CNEMC; https://air.cnemc.cn:18007/, last
access: 1 January, 2023) and Weather Underground (https://www.wunderground.com/, last access:
1 January, 2023), respectively. CNEMC operated 12 monitoring sites in Harbin, with 3 of them



located within ~5 km from the HIT sampling site. The reconstructed $PM_{2.5}$ masses, which were
derived from observational results on aerosol compositions at HIT, were generally in line with the
fine particle concentrations directly measured at the nearby CNEMC sites. Here the reconstructed
$PM_{2.5}$ was calculated as the sum of organic matter (1.6 ×OC), elemental carbon and inorganic ions.
Comparison of the reconstructed and directly-measured $PM_{2.5}$ concentrations showed relative
standard deviations of 9–11% (in terms of median value) for the three CNEMC sites nearby,
demonstrating HIT as a representative urban site for Harbin. In this study, only the air quality data
from the nearest CNEMC site, i.e., Taping Hongwei Park, were further investigated together with
the aerosol components measured at HIT.
**3. Results and discussion**
**3.1 Why was the wintertime brown carbon more absorbing at night?**
The wavelength-resolved $b_{abs}$ and MAE were primarily explored at 365 nm, and the
corresponding values were referred to as $(b_{abs})_{365}$ and $MAE_{365}$, respectively. $(b_{abs})_{365}$ and MSOC
correlated strongly for the winter campaign (Figure 1a), that the linear regression of $(b_{abs})_{365}$ against
MSOC led to an $r$ value of 0.97 and a slope of 1.63 ±0.02 $m^2/gC$ (with the intercept set as zero;
$MAE_{365}$ averaged 1.55 ± 0.18 $m^2/gC$). However, the nighttime samples were found to exhibit
slightly higher $MAE_{365}$ values than the daytime ones, with averages of 1.61 ±0.15 and 1.48 ±0.18
$m^2/gC$, respectively (Figure 1b). In this study, we did not perform source apportionment analysis
for brown carbon due to the relatively small number of samples collected. Instead, several indirect
indicators were introduced to interpret the diurnal variations of $MAE_{365}$.
The first indicator was the levoglucosan to OC ratio (LG/OC; on a basis of carbon mass, the
same hereinafter). In general, higher LG/OC values indicate a stronger contribution of biomass





burning (BB) emissions to OC. The BB activities in January could be attributed primarily to
household use of biofuels, e.g., for heating and cooking. This is because (i) few fire hotspot was
detected in Harbin and surrounding regions throughout the winter campaign (Figure 2a), and (ii) the
relationship between LG and water-soluble potassium ($K^+$), another commonly-used BB tracer, did
not show evidence for apparent influence of open burning (Figure 3a). As suggested by previous
studies conducted during heating season in Harbin (Cheng et al., 2022b), the LG to $K^+$ ratios were
relatively low and constant (~0.5) with the absence of agricultural fires, but became substantially
higher (typically above 1.0) during open burning episodes. This pattern was attributed to the
relatively low combustion efficiencies (CE) of agricultural fires, which favored the increase of LG
emissions but would not change $K^+$ emissions significantly (Gao et al., 2003). It should be noted
that in Cheng et al. (2022b), CE were not directly measured for different types of burning activities
and instead were investigated based on the ratios of BB organic carbon to BB elemental carbon ($R_{BB}$,
derived from positive matrix factorization, i.e., PMF, analysis). Substantial increases of $R_{BB}$ were
repeatedly observed during open burning episodes occurring in different seasons, e.g., winter or
spring depending on the regulatory policies. Thus the agricultural fires were inferred to have
relatively low CE levels (Cheng et al., 2022b), as BB source emission studies typically showed a
decreasing trend for the emission ratio of organic carbon to elemental carbon with increasing
combustion efficiency (Pokhrel et al., 2016; McClure et al., 2020). Actually, crops residues burned
on farmland were usually not intentionally dried and thus could have relatively high water contents.
This may partially explain the relatively low CE of agricultural fires. In the present study, LG
correlated strongly with $K^+$ for the entire January ($r = 0.96$, with a slope, i.e., $\Delta LG/\Delta K^+$, of $0.55 \pm$
$0.02$; Figure 3a) and the LG to $K^+$ ratios averaged $0.46 \pm 0.11$, pointing to the dominance of





residential burning in BB emissions. In addition, the residential burning activities were more
intensive at night, as can be seen from the elevated LG/OC compared to daytime results (1.10 ±
0.26% vs. 0.88 ±0.22%; Figure 1c). Comparison of the LG to EC ratios between the nighttime and
daytime samples (0.22 ±0.06 vs. 0.15 ±0.05) reached the same conclusion. Indeed, biomass burning
could emit a number of strong chromophores such as nitrogen-containing aromatic compounds
(Mohr et al., 2013; Lin et al., 2016, 2017; Xie et al., 2019; Salvador et al., 2021). However, for the
January samples, $MAE_{365}$ did not show clear dependence on LG/OC or LG/EC ($r = 0.42$ and 0.12,
respectively; Figure 1e), suggesting that in addition to BB emissions, there must exist other factors
that were more responsible for the diurnal variations of wintertime $MAE_{365}$.

The second indicator was $R_{S/N}$, defined as the ratio of (n-sulfur dioxide + n-sulfate) to (n-

nitrogen dioxide + n-nitrate), where "n" indicates molar concentration. Given that sulfate and nitrate
are typically considered as secondary, $R_{S/N}$ could be roughly traced back to the emission ratios of
sulfur dioxide ($SO_2$) to nitrogen oxides ($NO_x$), i.e., $E_{S/N}$, from combustion of various types of fuels
(e.g., coal, gasoline, diesel and biomass). Previous studies suggested that $E_{S/N}$ differed substantially
between emissions from vehicles, coal combustion and biomass burning. In China, the fuel quality
standards have been greatly strengthened for on-road vehicles since early 2000s, e.g., the maximum
sulfur content allowed in diesel was reduced from 2000 ppm (required by the China I standard
implemented in 2002) to 10 ppm (required by the China V standard implemented in 2017). Thus,
recent studies on vehicular exhausts typically suggested that the $SO_2$ emission factors (EF-$SO_2$)
were about two orders of magnitude lower than those of NOx (EF-NOx; Zhang et al., 2015; Li et
al., 2019) and consequently, the corresponding $E_{S/N}$ should be approximately $\sim 10^{-2}$. EF-$SO_2$ were
also usually lower than EF-NOx for biomass burning (Zhang et al., 2000; McMeeking et al., 2009;





Liu et al., 2016; Wu et al., 2022), but their differences were not as large as those observed in vehicle
emissions, leading to $E_{S/N}$ values of ~$10^{-1}$. Unlike vehicles or biomass burning, coal combustion
usually resulted in higher EF-$SO_2$ compared to EF-NOx (Zhang et al., 2000; Du et al., 2017; Li et
al., 2017), which could be translated to $E_{S/N}$ values of above one. On the other hand, primary species
could be transformed rapidly during atmospheric aging, e.g., a sharp loss of $NO_x$ and a
corresponding burst in nitrate were observed shortly after emission when tracking plumes from
diesel trucks (Shen et al., 2021) and agricultural fires (Akagi et al., 2012; Liu et al., 2016). Thus it
should be acceptable to assume that for the pollutants emitted by a specific source, the $R_{S/N}$ of aged
plumes was generally comparable with the $E_{S/N}$ of fresh emissions.

The ambient $R_{S/N}$ averaged $0.6 \pm 0.2$ during the winter campaign, differing substantially from

the $E_{S/N}$ of coal combustion or vehicle emissions but in the same order of magnitude as the $E_{S/N}$ of
biomass burning. Actually, no evidence supported BB emissions as a major regulating factor for
$R_{S/N}$, e.g., as indicated by the insignificant correlations between $R_{S/N}$ and LG/EC ($r = 0.24$ and $0.01$
for the daytime and nighttime samples, respectively). Then $R_{S/N}$ was expected to be more sensitive
to the changes of coal combustion and vehicle emissions, e.g., increase of coal combustion
emissions would effectively elevate $R_{S/N}$ whereas higher vehicle emissions favor the decrease of
$R_{S/N}$. During the winter campaign, lower $R_{S/N}$ were observed at night (Figure 1d), averaging $0.5 \pm$
$0.1$ compared to an average $R_{S/N}$ of $0.7 \pm 0.2$ for the daytime samples. In principle, this pattern could
be caused by decreased coal combustion emissions and/or increased vehicle emissions at night.
However, it seemed that the former did not play an important role, since many coal combustion
activities (e.g., those for heating supply, power generation and some industrial processes) were
uninterruptible, i.e., would not be stopped at night (Lian et al., 2020; Chu et al., 2021; Yuan et al.,



2021). Then the most likely cause for the lower nighttime $R_{S/N}$ was increased vehicle emissions.
According to the Road Traffic Regulations released by Harbin, heavy-duty diesel trucks (HDDT),
which are known to include high- or super-emitters (Dallmann et al., 2012), are allowed to run on
the roads in the main urban area only from 21:00 to 5:00 of the next day. This to a large extent
explains the inference on the increase of vehicle emissions during nighttime. $MAE_{365}$ exhibited a
clear negative dependence on $R_{S/N}$ for all the winter samples (Figure 1f), suggesting vehicle
emissions, especially those from HDDT, as a dominant influencing factor for $MAE_{365}$ (under the
precondition of relatively stable coal combustion emissions).

The last two indicators were associated with secondary aerosol formation, including the sulfur

oxidation ratio (SOR) and the nitrogen oxidation ratio (NOR) defined as n-sulfate/(n-sulfate + n-
$SO_2$) and n-nitrate/(n-nitrate + n-$NO_2$), respectively. The entire winter campaign experienced low
temperatures, which averaged −16 ± 5 and −21 ± 6 °C for the daytime and nighttime samples,
respectively. In general, the transformation of gaseous precursors to secondary inorganic ions was
inefficient in the frigid atmosphere, as indicated by the overall low levels of both SOR and NOR.
However, both indicators exhibited noticeable differences between daytime and nighttime samples.
The diurnal variation of SOR was found to be associated with the higher relative humidity (RH)
levels at night (Figure 4a). For the vast majority of winter samples, RH fell into the ranges of 60–
80 and 70–90% during daytime and nighttime, respectively. SOR were largely unchanged when RH
increased from 60–70% to 70–80% during the day, whereas for the common RH range shared by
the daytime and nighttime samples (i.e., 70–80%), SOR were slightly lower at night, likely due to
the drop of temperatures. In addition, a positive dependence of SOR on RH was evident for the
nighttime samples. Although SOR showed almost the same median values (~0.1) for the RH ranges



of 70–80 and 80–90% at night, relatively high SOR levels of above 0.2 were more frequently
observed in the latter case. Such high SOR were rarely seen during the day, indicating that RH
played a more important role than temperature in sulfate formation. The enhanced sulfate formation
at high RH was presumably through heterogeneous reactions (Su et al., 2020; Liu et al., 2021), since
the low temperatures encountered during the winter campaign did not rule out the presence of
aerosol water, e.g., liquid water was observed to remain super-cooled in clouds down to
temperatures of as low as –40 °C (Tabazadeh et al., 2002). Compared to SOR, different patterns of
diurnal variation were observed for NOR (Figure 4b). First, the difference between daytime and
nighttime NOR was more significant for the RH range of 70–80%, e.g., as indicated by the larger
decrease of median NOR at night (0.06, compared to a corresponding value of 0.02 for SOR).
Second, the nighttime NOR elevated substantially as RH increased from 70–80% to 80–90%, but
still with lower levels compared to the daytime results. Given that relatively low temperatures favor
the partitioning of semi-volatile nitrate into aerosol phase, the less efficient nitrate formation at night
could not be explained by the partitioning process and instead should be primarily attributed to
reduced photooxidation of $NO_2$ (Chen et al., 2020). Based on a synthesis of the diurnal variations
observed for SOR and NOR, the nighttime samples were characterized by enhanced heterogeneous
chemistry, which did not require sunlight as indicated by the RH-dependent increase of SOR under
dark conditions, and weakened photochemical reactions. The overall effect of these two factors on
secondary organic aerosol (SOA) formation was inconclusive and thus it remained difficult to
unfold the role of SOA in the diurnal variations of $MAE_{365}$. Actually, it appeared that $MAE_{365}$ was
not strongly influenced by SOA during the winter campaign. For example, when RH increased from
70–80% to 80–90% at night, the $MAE_{365}$ were nearly constant (e.g., with the same average value of



1.6 $m^2/gC$ for the two RH ranges) despite the enhancement of heterogeneous chemistry.
**3.2 Why did the springtime $MAE_{365}$ show more significant diurnal variations?**
Compared to the wintertime results, the average $MAE_{365}$ was lower in spring (1.33 vs. 1.55
$m^2/gC$) but the corresponding standard deviation was much higher (0.62 vs. 0.18 $m^2/gC$), indicating
that the spring samples varied more significantly with respect to the absorption capacity of brown
carbon (Figure 5a). This feature could also be seen from the more pronounced diurnal variations of
$MAE_{365}$ observed in spring (Figure 5b), e.g., the nighttime $MAE_{365}$ were on average ~70% and 10%
larger than the daytime values during the spring and winter measurement periods, respectively. For
the winter campaign, the slightly elevated $MAE_{365}$ at night had been primarily attributed to increased
vehicle emissions, as indicated by a ~35% decrease of $R_{S/N}$. In spring, $R_{S/N}$ were also lower at night,
by ~40% compared to the daytime results (Figure 5d). Given that the two campaigns showed
comparable discrepancies between the nighttime and daytime $R_{S/N}$, increase of vehicle emissions at
night was presumably not the dominant driver for the much stronger diurnal variations of $MAE_{365}$
observed in spring. Actually, $MAE_{365}$ was almost independent of $R_{S/N}$ for the spring samples. For
example, the $MAE_{365}$ values were found to fall into two well-separated ranges (above 2 and ~0.5–
1.5 $m^2/gC$, with the former observed only at night) for the samples with relatively low $R_{S/N}$ levels
(below 0.4), indicating that reduced $R_{S/N}$ was ineffective to explain the high $MAE_{365}$ events
encountered in spring (Figure 5f). In addition to increased vehicle emissions at night, therefore,
there must exist other factors which were more responsible for the significant diurnal variations of
springtime $MAE_{365}$.
We first evaluated the influence of secondary aerosol formation. The spring campaign
experienced lower RH and substantially higher temperatures compared to winter, by ~25% and





30 ℃, respectively. The springtime SOR were lower than the wintertime results (0.12 $\pm$ 0.06 vs.
0.15 $\pm$ 0.07), whereas an opposite pattern was observed for NOR (0.16 $\pm$ 0.08 vs. 0.12 $\pm$ 0.06). The
seasonal variations of SOR and NOR provided additional evidence for the inferences that the sulfate
and nitrate formation was more strongly contributed by heterogeneous and photochemical reactions,
respectively. For the spring campaign, the daytime and nighttime SOR were in general comparable
(Figure S1a) and no clear evidence was observed for the prevalence of heterogeneous chemistry,
presumably due to the rare occurrence of high RH conditions either during the day or at night.
Unlike SOR, the daytime NOR were considerably higher than the nighttime results (0.18 $\pm$ 0.09 vs.
0.14 $\pm$ 0.08; Figure S1b), pointing to enhanced photochemistry during the day. This pattern could
be partially responsible for the relatively low daytime $MAE_{365}$, since secondary brown carbon was
typically less light-absorbing than primary BrC (Kumar et al., 2018; Cappa et al., 2020; Ni et al.,
2021). However, $MAE_{365}$ did not exhibit clear dependence on NOR or the nitrate to OC ratio ($NO_3^-$
/OC), e.g., the high $MAE_{365}$ events were found to be associated with moderate NOR and $NO_3^-$/OC
levels (Figure S2). Thus for the spring campaign, photochemistry should not be the major
influencing factor for $MAE_{365}$, either.

We then investigated the role of biomass burning. Unlike the wintertime results, $MAE_{365}$

showed a strong positive correlation with LG/OC ($r$ = 0.84) in spring (Figure 5e), suggesting
biomass burning emissions as the dominant driver for the variations of $MAE_{365}$. It is noteworthy
that the LG to OC ratios were substantially higher in spring than in winter, with averages of 3.11 $\pm$
1.70% and 0.99 $\pm$ 0.26%, respectively. This pattern could not be explained by seasonal variations in
residential consumption of biofuels, since April experienced much higher temperatures than January
(averaging 11 and −19 ℃, respectively). Instead, the elevated springtime LG/OC should be





attributed primarily to open burning, as supported by the intensive fire hotspots detected around
Harbin in April (Figure 2b). The seasonal variations of LG to $K^+$ ratio (LG/$K^+$) also suggested that
the dominant burning ways were different between winter and spring. Compared to the relatively
small and constant LG/$K^+$ observed in January (0.46 ±0.11), the ratios were nearly tripled in April
(1.28 ±0.61) with more significant sample-by-sample differences (between ~0.5–3.5) (Figure 3b).
Recalling that the transition from flaming to smoldering combustion favored the increase of LG/$K^+$
(Gao et al., 2003), the springtime burning should have relatively low and variable combustion
efficiencies. This inference was in line with the fact that the agricultural fires were usually
uncontrolled, e.g., with respect to water content of crop residues and abundance of oxygen. In all,
for the spring campaign, the dominant driver for the variations of LG/OC and MAE$_{365}$ could be
further identified as open burning. Subsequently, the higher LG/OC and MAE$_{365}$ at night (Figures
5b–5c) could be attributed primarily to increased agricultural fires. The preference on nighttime
burning was not surprising, since the agricultural fires were illegal, i.e., nominally prohibited by the
Government of Heilongjiang Province.

It should be noted that the agricultural fire emissions increased LG/OC but had minimal

influence on $R_{S/N}$ (Figure S3). For example, the nighttime samples collected in spring differed
substantially with respect to the impact of agricultural fires, as indicated by their variable LG/OC
which spanned nearly one order of magnitude. However, no clear pattern was observed for $R_{S/N}$ with
increasing LG/OC, e.g., linear regression of $R_{S/N}$ on LG/OC showed an extremely low $r$ value of

0.07.

The frequent occurrences of agricultural fires during April, 2021 to some extent masked the

"background" MAE$_{365}$, i.e., the value representative for the spring conditions without significant



influence of open burning. In spring, all the samples with $LG/K^+$ ratios of above one, i.e., a chemical
signature for apparent impacts of agricultural fires, were found to have LG/OC ratios larger than
2%. Thus in the following discussions, LG/OC of > 2% was used as an indicator for open burning
episodes and correspondingly, spring samples with LG/OC of below 2% were referred to as typical
ones. $MAE_{365}$ averaged $0.80 \pm 0.22$ m²/gC for the typical samples of spring, lower than results from
the winter campaign ($1.55 \pm 0.18$ m²/gC; Figure S4a). This seasonal pattern coincided with the
overall lower $R_{S/N}$ in spring (Figure S4b). It was unlikely that the number of in-use vehicles or the
fleet composition in Harbin could vary significantly between January and April of the same year.
Thus the reduced springtime $R_{S/N}$, i.e., the relatively low $MAE_{365}$ with the absence of agricultural
fires, should be caused mainly by the decrease of coal combustion emissions, e.g., due to the less
demand for heating.

**3.3 Unique wavelength dependence of BrC absorption during agricultural fire episodes**

The agricultural fires not only elevated $MAE_{365}$ but also changed the wavelength dependence
of brown carbon. For the wavelength range used for AAE calculation (310–460 nm), the detection
limit of optical attenuation ($ATN_{LOD}$) was ~0.02, which was determined as three times the maximum
standard deviation of parallel $ATN_\lambda$ results from blank filters. Before further discussions, we
introduced a new term "relative $\ln(ATN_\lambda)$", i.e., $\ln(ATN_\lambda)^*$ calculated as $\ln(ATN_\lambda) - \ln(ATN_{LOD})$.
A benefit of using the new term was that a $\ln(ATN_\lambda)^*$ value of zero corresponded to $ATN_{LOD}$ and
thus could be interpreted independently, e.g., $ATN_{LOD}$ was independent of the sampling or analytical
procedures such as the volume of methanol used for extraction. It should be noted that the use of
$\ln(ATN_\lambda)^*$ would not influence the determination of AAE, since the same slope would be derived
from the regressions of $\ln(ATN_\lambda)^*$ and $\ln(ATN_\lambda)$ on $\ln(\lambda)$. For the typical samples of spring, the



dependence of $\ln(\text{ATN}_\lambda)^*$ on $\ln(\lambda)$ could be properly approximated by a linear function, usually with
$r$ values of above 0.995. In this case, AAE could be reliably determined, and an average value of
6.92 ±0.28 was obtained.

The relationship between $\ln(\text{ATN}_\lambda)^*$ and $\ln(\lambda)$ became non-linear for the open burning episodes.

To more quantitatively describe the non-linearity, we added an "auxiliary line" to each measured
spectrum (Figure 6a), by drawing a line between the two points with $x$ values of $\ln(310)$ and $\ln(460)$.
The "auxiliary line" could be considered an assumed spectrum with linear dependence of $\ln(\text{ATN}_\lambda)^*$
on $\ln(\lambda)$. The measured spectrum was always above the assumed one and their largest difference
was typically observed at ~365 nm, pointing to the presence of distinct BrC chromophores with
absorption peak around this wavelength.

The influence of such chromophores on BrC absorption could be estimated by the following

three indicators. The first one ($F$) was related to the difference between the measured and assumed
$\ln(\text{ATN}_\lambda)^*$ at 365 nm:
$F = \left[ \ln\left(\text{ATN}_{365}\right)^*_{\text{m}} - \ln\left(\text{ATN}_{365}\right)^*_{\text{a}} \right] \Big/ \ln\left(\text{ATN}_{365}\right)^*_{\text{a}}$ , where the subscripts "m" and "a" indicate
results from the measured and assumed spectra, respectively (Figure 6a). The second indicator ($K$)
was related to the area enclosed between the two spectra ($S_2$): $K = S_2/S_1$, where $S_1$ indicates the area
enclosed by the assumed spectrum and $x$-axis (Figure 6b). The last indicator was $\Delta(b_{\text{abs}})_{365}$
calculated as $\left(b_{\text{abs}}\right)^{\text{m}}_{365} - \left(b_{\text{abs}}\right)^{\text{a}}_{365}$ , where the superscripts "m" and "a" indicate absorption
coefficients calculated based on the measured and assumed spectra, respectively. $F$ and $K$ exhibited
a strong linear correlation for the open burning episodes ($r = 0.99$; Figure 6c), indicating that the
differences between the measured and assumed spectra were likely caused by the same class of BrC
compounds. In addition, these compounds could be primarily traced back to biomass burning, since



$\Delta(b_{abs})_{365}$ showed a positive dependence on LG/OC (Figure 6d). A candidate for such compounds
was $C_7H_7NO_4$ (a methyl-substituted nitrocatechol), based on a synthesis of absorption spectra
measured for various BrC chromophores (Huang et al., 2020) and molecular characterization results
for biomass burning emissions (Lin et al., 2016, 2017; Xie et al., 2019, 2020). Chamber experiments
by Iinuma et al. (2010) suggested that $C_7H_7NO_4$ could also be formed through photooxidation of
gaseous precursors emitted by biomass burning (*m*-cresol). In this study, however, all the samples
with relatively high $\Delta(b_{abs})_{365}$ levels (e.g., above 20 Mm$^{-1}$) were collected at night, indicating that
the distinct BrC chromophores with absorption peak at ~365 nm (like $C_7H_7NO_4$) were more strongly
associated with primary emissions from agricultural fires. In addition, the chromophores seemed to
be subject to photo-bleaching, as both *F* and *K* decreased substantially (by ~65%) during the day
compared to the nighttime results (Figure 7).

For the open burning episodes, the distinct absorption peak at ~365 nm prohibited a proper

determination of AAE. If enforcing a linear function for the dependence of $\ln(ATN_\lambda)^*$ on $\ln(\lambda)$, lower
*r* values would be derived (down to ~0.97, with an average of 0.992 $\pm$0.007) compared to the typical
samples (averaging 0.998 $\pm$0.002). In addition, *r* showed a decreasing trend with the increase of
LG/OC (Figure 6e), suggesting that the relationship between $\ln(ATN_\lambda)^*$ and $\ln(\lambda)$ deviated more
significantly from linearity as the ~365 nm absorption peak, i.e., the influence of agricultural fires,
became more significant. We suggest that for the open burning episodes, the AAE results should be
interpreted with caution, although they could be calculated mathematically with reasonable *r* values
(e.g., even the minimum *r* appeared acceptable).
**3.4 Diurnal variations of wintertime AAE**

Similar to the typical samples of spring, $\ln(ATN_\lambda)^*$ exhibited linear dependences on $\ln(\lambda)$ for





all the winter samples. The wintertime AAE were higher at night compared to those observed during
the day (with averages of $7.33 \pm 0.14$ and $6.76 \pm 0.11$, respectively), consistent with the pattern
observed during winter in Beijing (Li et al., 2020). The relative abundance of secondary OC (SOC)
has been considered an important influencing factor for AAE, e.g., an increasing trend was observed
for AAE during long-range transport of BrC over the Indo-Gangetic Plain (Dasari et al., 2019).
Although SOC or its organic tracer was not determined in this study, previous source apportionment
results from Harbin (based on PMF) showed a strong correlation between SOC and sulfate, with
largely consistent relationships among different campaigns (Cheng et al., 2022b). Thus we used
sulfate as an indicator for SOC. During the winter campaign, the sulfate to OC ratios were lower at
night (averaging 0.38, compared to 0.44 during the day), pointing to decreased fractions of SOC in
OC. This inference was consistent with the higher LG/OC and $R_{S/N}$ levels observed at night, which
had been attributed to increased emissions from residential biomass burning and vehicular exhausts,
respectively. Thus regarding the association between AAE and SOC formation, results from the
winter campaign were inconsistent with Dasari et al. (2019), but the reason remained unclear.
Molecular characterization of organic aerosols should be necessary to unfold the response of AAE
to changes in BrC sources.
**4. Conclusions**
Diurnal variations of BrC were investigated during two distinct seasons in the northernmost
megacity in China. The winter campaign was characterized by low temperatures rarely seen in other
hotspots of air pollution studies such as the North China Plain. The wintertime BrC aerosols were
slightly more absorbing at night, with an average $MAE_{365}$ of $1.61 \pm 0.15$ m$^2$/gC compared to $1.48 \pm$
$0.18$ m$^2$/gC during the day. Various indicators were used to explain the observed diurnal variations





of MAE$_{365}$, including those associated with biomass burning emissions (LG/K$^+$ and LG/OC),
relative importance of coal combustion and vehicle emissions ($R_{S/N}$) and secondary aerosol
formation (SOR and NOR). For the winter campaign, the nighttime samples were characterized by
increased BB emissions from residential sources, enhanced heterogeneous chemistry and weakened
photochemical reactions. But none of these factors was identified as the dominant driver for the
higher MAE$_{365}$ at night. Instead, MAE$_{365}$ exhibited a negative dependence on $R_{S/N}$, and the lower
$R_{S/N}$ and thus higher MAE$_{365}$ at night were primarily attributed to increased emissions from heavy-
duty diesel trucks, which were not allowed for the main urban area during the day. In addition, the
wintertime AAE were higher at night but it remained difficult to unfold the underlying connection
between this diurnal pattern and the changes in BrC sources.

The spring campaign was characterized by frequent occurrences of agricultural fires, with more

pronounced diurnal variations of MAE$_{365}$ (averaging $0.98 \pm 0.31$ and $1.69 \pm 0.65$ m$^2$/gC for the
daytime and nighttime samples, respectively). Unlike winter, the springtime MAE$_{365}$ were mainly
influenced by open burning emissions, as suggested by the positive dependence of MAE$_{365}$ on
LG/OC and the lack of correlation between MAE$_{365}$ and $R_{S/N}$. The higher nighttime LG/OC
indicated that the farmers preferred burning the crop residues at night, presumably because
agricultural fires were nominally prohibited by the local government. In addition, BrC exhibited
distinct light absorption spectra during agricultural fire episodes, as indicated by the non-linear
relationship between ln(ATN$_\lambda$)$^*$ on ln($\lambda$). The non-linearity was mainly caused by chromophores
with absorption peak at ~365 nm, which became more significant with increasing BB influence. A
candidate for the compounds at play was C$_7$H$_7$NO$_4$, based on a synthesis of absorption spectra
measured for various BrC chromophores and molecular measurement results for BB emissions. The



presence of such chromophores, i.e., the distinct absorption peak at ~365 nm, prohibited a proper
determination of AAE for the spring samples impacted by agricultural fires.
**Data availability.** Data described in this manuscript can be accessed at
https://doi.org/10.5281/zenodo.7590785 (Cheng, 2023).
**Author contributions.** YC and JL designed the study and prepared the paper with inputs from all
the co-authors. XC, YZ and QY carried out the experiments. QZ and KH validated the results and
supervised the study.
**Competing interests.** Author Qiang Zhang is a member of the editorial board of *Atmospheric*
*Chemistry and Physics*. The peer-review process was guided by an independent editor, and the
authors have also no other competing interests to declare.
**Acknowledgements.** The authors thank Zhen-yu Du at National Research Center for Environmental
Analysis and Measurement, and Lin-lin Liang at Chinese Academy of Meteorological Sciences for
their help in sample analysis.
**Financial support.** This research has been supported by the National Natural Science Foundation
of China (42222706), the Natural Science Foundation of Heilongjiang Province (LH2020D011),
Fundamental Research Funds for the Central Universities, and Heilongjiang Touyan Team.

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



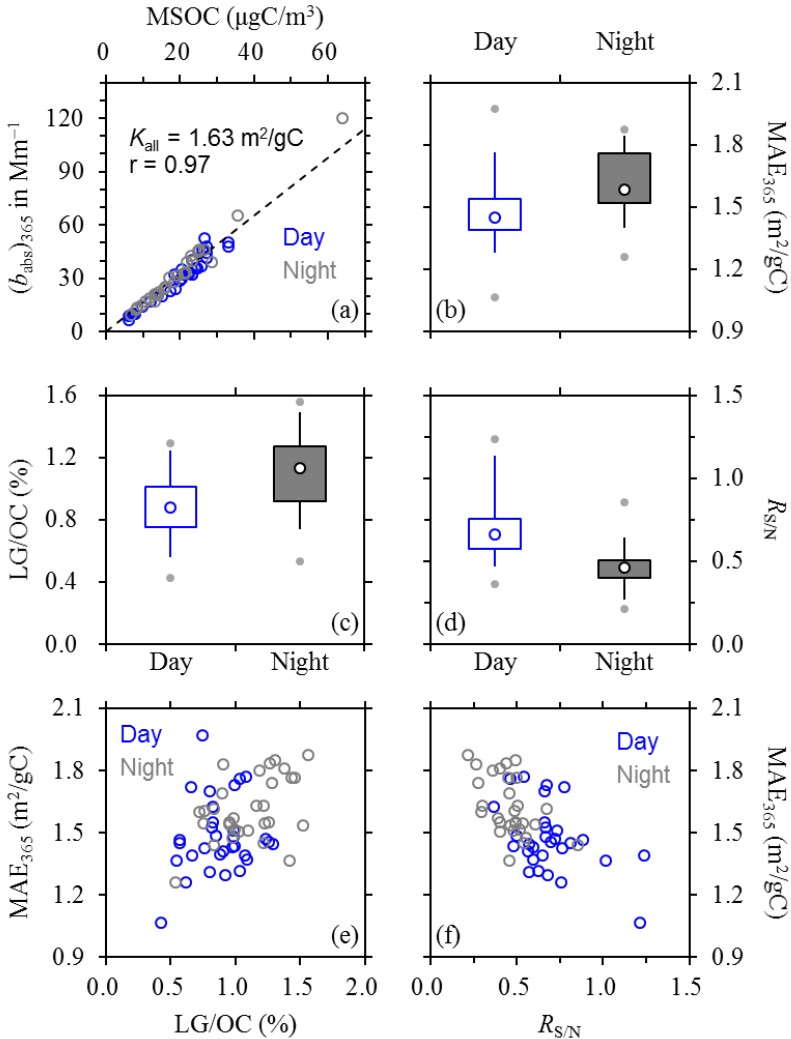

**Figure 1.** (a) Dependence of $(b_{abs})_{365}$ on MSOC, (b–d) diurnal variations of $MAE_{365}$, LG/OC (on a basis of carbon mass) and $R_{S/N}$, and (e–f) dependences of $MAE_{365}$ on LG/OC or $R_{S/N}$ during winter. In (a), the dashed line indicates linear regression result based on all the winter samples, with $K_{all}$ as slope (intercept was set as zero). In (b-d), lower and upper box bounds indicate the 25th and 75th percentiles, the whiskers below and above the box indicate the 5th and 95th percentiles, the solid circles below and above the box indicate the minimum and maximum, and the open circle within the box marks the median (the same hereinafter). Comparison of (e) and (f) suggests that the wintertime $MAE_{365}$ was more strongly influenced by $R_{S/N}$ compared to LG/OC. The dependence shown in (f) could be approximated by the following function for all the winter samples ($r = 0.61$): $MAE_{365} = (-0.51 \pm 0.09) \times R_{S/N} + (1.84 \pm 0.05)$.



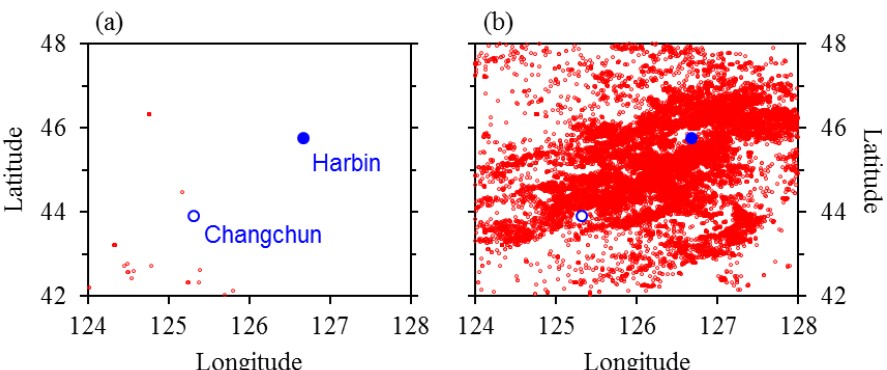

**Figure 2.** Active fires (red circles) detected during the **(a)** winter and **(b)** spring measurement periods around Harbin. The location of another central city of the HC metropolitan area, Changchun, is also shown. The fire data were based on the joint NASA/NOAA Suomi National Polar-orbiting Partnership (S-NPP) satellite, and were downloaded from the Fire Information for Resource Management System (FIRMS; https://firms.modaps.eosdis.nasa.gov/, last access: 1 January, 2023).



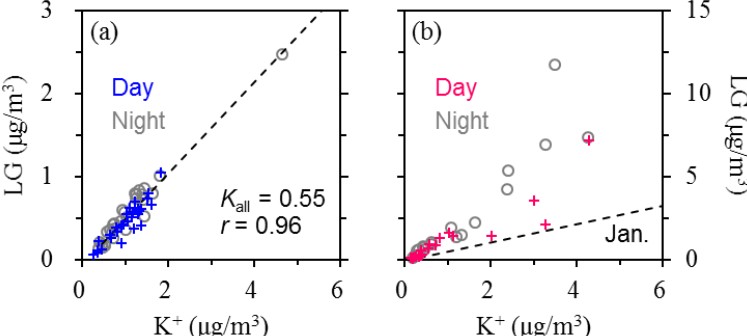

**Figure 3.** Dependences of levoglucosan on $K^+$ during **(a)** winter and **(b)** spring. In (a), the dashed line indicates linear regression result based on all the winter samples, with $K_{all}$ as slope. The regression line of winter campaign is also shown in (b) for comparison to highlight the increased and variable $LG/K^+$ ratios in spring. The relatively low and constant $LG/K^+$ in winter were attributed to residential burning of crop residues, a routine activity occurring every day in rural areas for cooking and heating. The higher $LG/K^+$ in spring were associated with agricultural fires, as supported by the intensive fire hotspots detected.



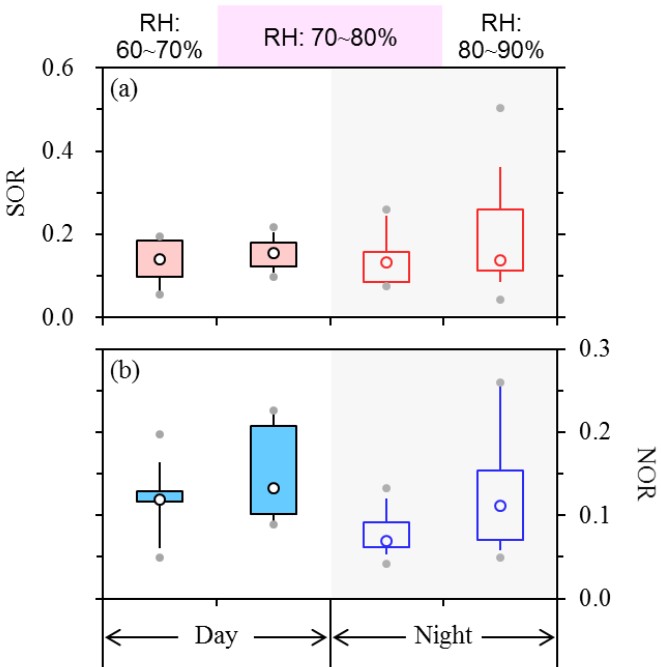

750

**Figure 4.** Diurnal variations of **(a)** SOR and **(b)** NOR in winter, with results from different RH
ranges shown separately. Daytime and nighttime samples had the same RH range of 70–80%,
whereas low RH levels of 60–70% and high RH levels of 80–90% occurred only during the day and
at night, respectively.

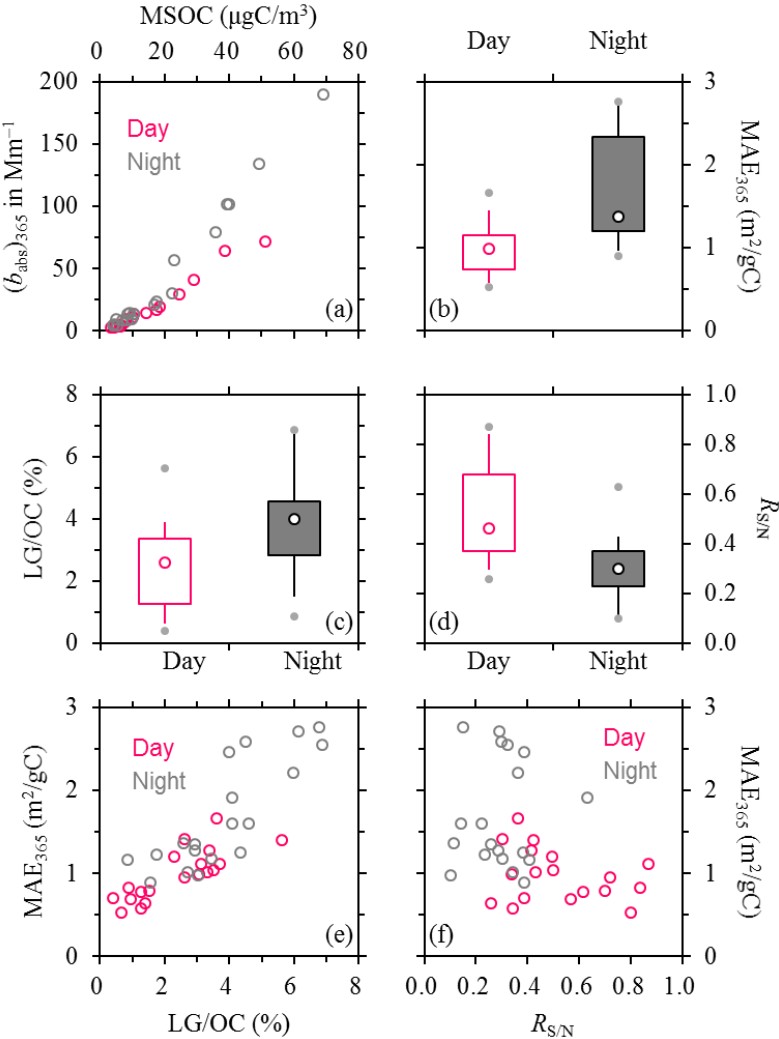

**Figure 5.** The same as Figure 1 but for spring. $MAE_{365}$ showed more pronounced diurnal variations in spring than winter, although the daytime vs. nighttime discrepancies in $R_{S/N}$ were comparable between the two seasons. Comparison of (e) and (f) suggests that unlike winter, the springtime $MAE_{365}$ was more strongly influenced by LG/OC than by $R_{S/N}$. The dependence shown in (e) could be approximated by the following function for all the spring samples ($r = 0.84$): $MAE_{365} = (30.48 \pm 3.28) \times LG/OC + (0.39 \pm 0.12)$.



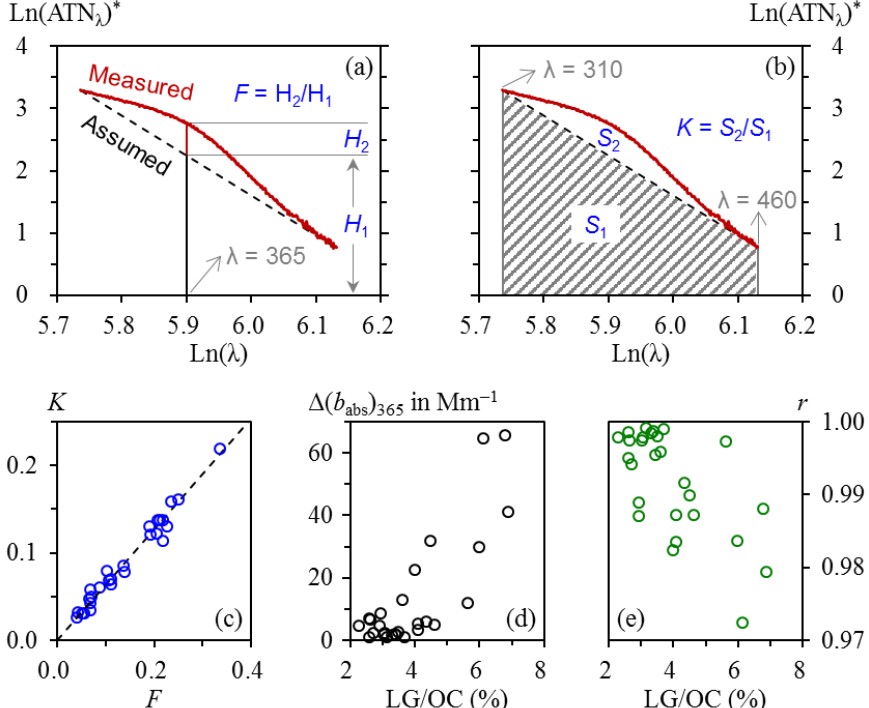

**Figure 6.** Nonlinearity of $\ln(\text{ATN}_\lambda)^*$ on $\ln(\lambda)$ during agricultural fire episodes in spring: **(a–b)** illustrations of the determination of $F$ and $K$, **(c)** comparison of $K$ and $F$, and **(d–e)** dependences of $\Delta(b_{\text{abs}})_{365}$ and $r$ on LG/OC. In (a) and (b), the measured spectrum correspond to the nighttime sample collected on April 21, 2021, which had an LG/OC of 6.87%; the assumed spectrum was generated by drawing a line between the two points with $x$ values of $\ln(310)$ and $\ln(460)$; $H_1$ indicates $\ln(\text{ATN}_{365})^*$ of the assumed spectrum, while $H_2$ indicates the difference in $\ln(\text{ATN}_{365})^*$ between the two spectra; $S_1$ indicates the area enclosed by the assumed spectrum and the $x$-axis, while $S_2$ indicates the area enclosed between the two spectra. In (c), the dashed line indicates linear regression result (intercept was set as zero) and the corresponding $r$ value was 0.99. In (e), $r$ was derived from linear regression of $\ln(\text{ATN}_\lambda)^*$ on $\ln(\lambda)$. Although the $r$ values seemed reasonable, the AAE results should be interpreted with caution given the apparent absorption peak at ~365 nm.





774

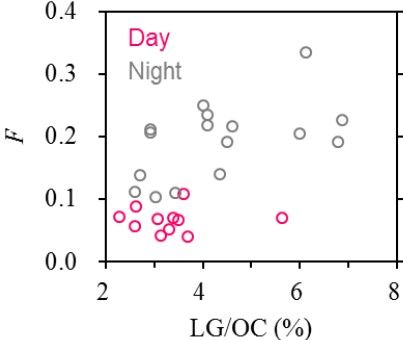

**Figure 7.** Dependence of $F$, a measure of the significance of the ~365 nm absorption peak, on
LG/OC during agricultural fire episodes in spring. For a given LG/OC range, $F$ decreased
substantially during the day, likely due to photo-blanching of chromophores associated with the
~365 nm peak. The same conclusion could be reached based on $K$, another indicator for the
significance of the ~365 nm peak.