# Peer review of "Measurement report: Diurnal variations of brown carbon during two distinct seasons in a megacity in Northeast China"

_Atmospheric Chemistry and Physics, 2023_

## Author Comment (AC1)

*General comments*

This manuscript investigated diurnal variations of BrC in a megacity in Northeast China. The studied region is distinct, because it has quite different meteorological conditions and emission sources compared to well-known hotspots such as Beijing and surrounding regions. So far, however, aerosols in this region remained poorly understood with limited studies, e.g., regarding their chemical compositions, physical properties, sources and impacts. In this context, the authors conducted field measurements during two distinct seasons in a "largely unexplored" megacity in Northeast China, and traced the diurnal variations of BrC back to the changes of aerosol sources. The unique light absorption spectra of BrC observed during open burning episodes are especially interesting. Therefore, I think this manuscript represents a valuable contribution to better understanding of haze pollution in Northeast China, and my overall assessment is that it could be considered for publication in ACP after addressing the comments below.

*Major point*

MAE, which can be converted to the imaginary part of the complex refractive index of BrC, is an important parameter for climate models. In addition to a summary of the observational results, implications of this study should also be involved in the Conclusions section. To my understanding, although the winter is much colder in Northeast China compared to Beijing, MAE did not show apparent difference between these two regions. This is a potentially important point for the spatial distribution of MAE, but was completely ignored by the authors. In addition, the authors should make clear recommendations regarding whether the diurnal variations of MAE need to be considered in climate models.

**Our responses:** We agree with the reviewer that implications of our results should be clearly stated. Actually, this point was also raised by the handling editor. Thus a new section entitled "Implications" was incorporated into the revised manuscript **(see lines 472-490)**:

*$MAE_{365}$ and AAE are key parameters for simulating climate effects of brown carbon. In winter, although Harbin experiences low temperatures rarely seen in other Chinese megacities, the observed $MAE_{365}$ and AAE were largely comparable with the typical results from other regions in Northern China (e.g., Beijing; Cheng et al., 2016). In addition, BrC's optical properties were indeed different between daytime and*

*nighttime samples, which were likely associated with increased HDDT emissions at night. However, the diurnal variations (~10% higher at night for both MAE$_{365}$ and AAE) appeared negligible compared to uncertainties in simulating the mass concentration of BrC, i.e., organic aerosol. Thus for typical winter conditions in Northern China (without open burning), it may be practical to use fixed MAE$_{365}$ and AAE values for estimating the wavelength-resolved absorption by organic aerosol in climate models.*

*The spring campaign suggested another scenario, that the agricultural fires exhibited strong influences on optical properties of brown carbon, as highlighted by the ~365 nm peak in BrC's absorption spectra. The distinct peak on one hand effectively elevated MAE$_{365}$, and on the other hand complicated the determination of AAE. In addition, the peak became less significant during the day, indicating that the organic compounds at play were likely subject to photo-bleaching. BrC emitted by the fires remained difficult to constrain, partially due to the variable combustion efficiencies. This in turn resulted in challenges for simulating climate effects of the open burning aerosols. Given the massive agricultural sector in Northeast China, more studies are necessary to understand the emissions, transformation and impacts of the fire-induced pollutants.*

***Specific Points***

**(1)** Lines 34-35. Suggest towing down this statement.

**Our responses:** The sentence was changed to "*The presence of the ~365 nm peak complicated the determination of absorption Ångström exponents for the agricultural fire-impacted samples*" **(see lines 35-38)**.

**(2)** Line 75. Typical temperatures during winter in Beijing should also be given for comparison.

**Our responses:** Typical wintertime temperature was provided for Beijing (~0 ℃) as suggested **(see line 77)**.

**(3)** Line 96. Why did the daytime and nighttime samples have different sampling durations?

**Our responses:** The daytime and nighttime samples would have the same sampling duration (i.e., 8 hours) if the collection of the daytime samples was stopped at 17:00. However, the sunset time was typically between 16:00 and 17:00 during January in Harbin. Thus for the daytime samples, the sampling was ended at 16:00 and

consequently, their sampling duration (7 hours) was 1 hour shorter compared to the nighttime ones. Considerations in determining the starting and ending time were briefly explained in the revised manuscript: "*To avoid rush hours and considering the relatively early sunset time in winter (~16:00–17:00), daytime and nighttime samples were collected from 9:00 to 16:00 and from 21:00 to 5:00 of the next day, respectively*" **(see lines 97-98)**.

**(4)** Line 112. I think sonication could increase the extraction efficiency of BrC.

**Our responses:** We found that ultrasonicated and un-ultrasonicated methanol extracts showed comparable BrC absorption coefficients for ambient samples collected in Beijing. The comparison results were shown in Cheng et al. (2016), and we have provided this reference in the text for the extraction procedures **(see lines 113-114)**. To minimize the loss of insoluble carbon (e.g., EC), sonication was not applied in the present study when extracting the samples by methanol.

**(5)** Line 185. Suggest adding "compared to residential burning of crop residues" after "levels".

**Our responses:** The change was made as suggested **(see line 192)**.

**(6)** Line 277. Suggest adding "robustly" before "unfold".

**Our responses:** The change was made as suggested **(see line 285)**.

**(7)** Line 348. Is it necessary to introduce another indicator for open burning episodes?

**Our responses:** Here we introduced LG/OC because it could be used to not only identify open burning episodes but also estimate the strength of biomass burning impact. In other words, compared to LG/$K^+$, LG/OC is more suitable to be used to link the observed BrC optical properties with the influences of agricultural fires **(see lines 357-358)**.

**(8)** Line 363. This sentence is unclear, rewrite it.

**Our responses:** This sentence was rewritten as "*A benefit of using the new term was that a ln(ATN$_\lambda$)\* value of zero corresponded to ATN$_\lambda$ = ATN$_{LOD}$ and thus ln(ATN$_\lambda$)\* could be considered "real" absorption by chromophores in solutions.*" **(see lines 373-377)**.

**(9)** Lines 405-407. I would like to see a scatter plot showing the dependence of *r* on *F* or *K*.

**Our responses:** The plot was provided as suggested **(see Figure S5 and line 417)**.

[Figure]

**Figure R1.** Dependence of $r$ on $F$ during agricultural fire episodes in spring. $r$ was determined by regressing $\ln(ATN_\lambda)^*$ on $\ln(\lambda)$, while $F$ was a measure of the significance of the ~365 nm absorption peak. $r$ showed a clear decreasing trend with the increase of $F$. The same trend was observed when plotting $r$ against $K$, another indicator for the significance of the ~365 nm absorption peak. This figure was presented as Figure S5 in the revised manuscript.

**(10)** Line 738. Suggest changing "another" to "the other".

**Our responses:** The sentence was re-written as "*the HC metropolitan area has two central cities as marked by the blue circles*" in the revised manuscript **(see lines 771-772)**.

**(11)** Line 752. Suggest changing "the same" to "a common".

**Our responses:** The change was made as suggested **(see line 786)**.

**(12)** Line 761. I think it is better to clarify again that LG/OC involved in the equation was on a basis of carbon mass and was in %.

**Our responses:** The change was made as suggested **(see line 795)**.

**(13)** Table S2. Re-write the note as: AAE were not provided due to the frequent occurrences of agricultural fires, which could result in distinct peak at ~365 nm for the light absorption spectra of BrC.

**Our responses:** The change was made as suggested **(see Table S3)**.

---

## Author Comment (AC2)

*General comments*

The manuscript of Cheng et al. reports the diurnal variations of brown carbon (BrC) investigated during two distinct seasons in the northernmost megacity of China. Authors discussed drivers of diurnal BrC variations observed in two seasons, i.e., a cold winter (January 2021) and an agricultural fire-impacted spring (April 2021), relying on indicators of various sources.

This paper is well written, the experimental part is well presented and, along with citing the relevant literature, the experimental approach is well described. However, my main concern is directed to data presentation, interpretation, and drawing the conclusions as will be indicated later. Considering the importance of the topic that is the focus of this article, my overall assessment is that this paper should be considered for publication in ACP, but after major revision since there are some issues that need to be addressed to improve this work.

*Major point*

The authors hypothesized on more absorbing BrC at night, based on comparison of mean nighttime and daytime $MAE_{365}$ values in winter. However, I do not see that this difference is statistically significant. Furthermore, authors attempted to explain the drivers of observed "diurnal variations", but have not reached a clear conclusion, which is not surprising since it is double if the diurnal difference even exists. In fact, authors discussed that the predominant influencing factor for $MAE_{365}$ is vehicle emissions, especially those from nighttime HDDTA transport, based on the lower average $R_{S/N}$ observed at night (0.5 ±0.1) compared to $R_{S/N}$ for the daytime samples (0.7 ±0.2). The problem here is again that the average $R_{S/N}$ values obtained for the nighttime and daytime samples were not statistically different and such a conclusion is overstated.

The authors should first test the statistical significance of the $MAE_{365}$ difference between night and day in winter. Furthermore, the discussion and conclusions should be based on statistically reliable data, and rigorous arguments need to be added to this paragraph. I suggest rewriting this paragraph, including changing the title.

Diurnal variations of $MAE_{365}$ in spring (averaging 0.98 ± 0.31 and 1.69 ± 0.65 m²/gC) should also be disused based on statistically proven difference between the day and night samples.

**Our responses:** We agree with the reviewer that statistical analyses should be

performed to support comparisons involved in the manuscript. Thus we conducted $t$-tests and confirmed that: (i) for the winter campaign, the diurnal variations were statistically significant at the 95% confidence level for both $MAE_{365}$ and $R_{S/N}$ ($p = 0.004$ and 0.000, respectively), and (ii) the diurnal variations of $MAE_{365}$ and LG/OC ($p = 0.000$) were also statistically significant in spring. In the revised manuscript, $t$-test results were provided alongside descriptions of diurnal or seasonal differences, and were also summarized in a supplementary table. Based on the statistical results, we on one hand confirmed that major conclusions of the original manuscript still held, and on the other hand avoided overstatement **(see lines 167-170, 173-174, 201-203, 239, 263, 295-296, 299, 311, 318, 329, 425, 433-434, and Table S1)**.

**Table S1.** Summary of $t$-test results for the comparisons involved in the main text. A $p$ value of below 0.05 indicates statistically significant difference at the 95% confidence level.

| Compared parameters | $p$ value of $t$-test | Indication |
|---|---|---|
| *Winter campaign* | | |
| Daytime and nighttime $MAE_{365}$ | 0.004 | More absorbing BrC at night |
| Daytime and nighttime LG/OC (LG/EC) | 0.001 (0.000) | Increased residential burning emissions at night |
| Daytime and nighttime $R_{S/N}$ | 0.000 | Increased vehicle emissions at night |
| Daytime and nighttime SOR in the RH range of 70–80% | 0.417 | Relatively weak influence of photochemistry on sulfate formation |
| Daytime and nighttime NOR in the RH range of 70–80% | 0.005 | Relatively strong influence of photochemistry on nitrate formation |
| Daytime and nighttime AAE | 0.000 | Stronger wavelength dependence of BrC absorption at night |
| Daytime and nighttime sulfate/OC | 0.011 | Decreased SOC/OC ratios at night |
| *Spring campaign* | | |
| Daytime and nighttime $MAE_{365}$ | 0.000 | More absorbing BrC at night |

| | | |
|---|---|---|
| Daytime and nighttime LG/OC | 0.006 | Increased agricultural fire emissions at night |
| Daytime and nighttime $R_{S/N}$ | 0.000 | Increased vehicle emissions at night |
| Daytime and nighttime SOR | 0.489 | Insignificant diurnal variations of sulfate formation |
| Daytime and nighttime NOR | 0.083 | Insignificant diurnal variations of nitrate formation |
| $r$ values for typical samples and open burning episodes [derived from linear regression of $\ln(ATN_\lambda)^*$ on $\ln(\lambda)$] | 0.000 | Agricultural fire-induced non-linearity for BrC's absorption spectra shown on ln-ln scale |
| *Inter-campaign* | | |
| LG/K$^+$ in winter and spring | 0.000 | Different biomass burning ways in the two seasons (i.e., residential and open burning, respectively) |
| LG/OC in winter and spring | 0.000 | Stronger impacts of biomass burning in spring |
| SOR in winter and spring | 0.050 | —— |
| NOR in winter and spring | 0.012 | Significant seasonal variations of nitrate formation |
| Wintertime $MAE_{365}$ and $MAE_{365}$ of typical samples in spring | 0.000 | Less absorbing BrC in spring with the absence of agricultural fires |

*Specific points*

(1) Lines 240-241. $MAE_{365}$ exhibited a negative dependence on $R_{S/N}$ for nighttime samples? Please explain.

**Our responses:** Yes, $MAE_{365}$ exhibited a negative dependence on $R_{S/N}$ for the nighttime samples in winter, and their relationship [$MAE_{365} = (-0.57 \pm 0.18) \times R_{S/N} + (1.88 \pm 0.09)$; $r = 0.51$] was comparable with that derived from all the winter samples [$MAE_{365} = (-0.51 \pm 0.09) \times R_{S/N} + (1.84 \pm 0.05)$; $r = 0.61$]. The similar negative correlations suggested that the variation of nighttime $R_{S/N}$ might also be caused by the difference in numbers and/or emissions of heavy-duty diesel vehicles.

**(2)** Line 325. Please explain how Fig 2b is created. Are there cumulative fire events present for January and April? Please indicate this in the figure caption.

**Our responses:** Figure 2b was created using latitudes and longitudes of fire hotspots detected throughout the spring campaign. Figure 2a was created similarly. Both figures indicate cumulative fire events. This point was clarified in the revised manuscript **(see lines 769-771)**.

**(3)** Line 335. Is there any evidence of more frequent/intense nighttime burning from NASA/NOAA Suomi National Polar-orbiting Partnership (S-NPP) satellite, and/or the Fire Information for Resource Management System?

**Our responses:** The S-NPP satellite passed over Northeast China twice a day, approximately at noon and midnight, respectively. The fire hotspots were mainly detected during the day. However, this does not conflict with our inference on the prevalence of nighttime fires, which resulted in relatively high $LG/K^+$ levels compared to the daytime fires (1.73 $\pm$ 0.53 vs. 1.27 $\pm$ 0.35, $p = 0.018$). It had been observed that the transition from flaming to smoldering combustion favored the increase of $LG/K^+$ (Gao et al., 2003), thus the nighttime fires should have relatively low combustion efficiencies and consequently, they were more difficult to be detected by satellites. Cheng et al. (2021) found that the CMAQ air quality model significantly under-predicted OC and $PM_{2.5}$ during low-efficiency fire events, mainly due to the underestimation of open burning emissions by satellite-based inventory. Thus, we think compared to fire hotspots, directly-measured chemical signatures (e.g., $LG/K^+$ and LG/OC) could reflect the differences between daytime and nighttime fires more reliably.

**(4)** Lines 390-393. I agree that aromatic compounds with nitro-functional groups are good representatives of BrC related to biomass burning emission. I suggest not referring specifically to methylnitrocatechols, but rather to aromatic compounds with nitro-functional groups in general.

**Our responses:** Changes were made as suggested, i.e., "$C_7H_7NO_4$" mentioned throughout the manuscript were replaced by "aromatic compounds with nitro-functional groups" **(see lines 34-35, 401-403, 406 and 410)**.

**(5)** Lines 439-441. Based on my general comment above, please rewrite this part of the conclusion about the higher $MAE_{365}$ observed at night in winter samples.

**Our responses:** As mentioned in our responses to the major comment, the diurnal

variations were statistically significant at the 95% confidence level for both the wintertime $MAE_{365}$ and $R_{S/N}$ ($p$ = 0.004 and 0.000, respectively). Thus we think it should be acceptable to conclude that $MAE_{365}$ were higher at night, accompanied with increased nighttime $R_{S/N}$.

**(6)** L453-455. Please rewrite the sentence since in its current form one could read that your work also involves chromophore absorption spectra and molecular measurements.

**Our responses:** This sentence was re-written as "*Aromatic species with nitro-functional groups were a possible class of compounds that were at play*" **(see lines 465-468)**.

**Reference**

Cheng Y., Yu Q.Q., Liu J.M., Zhu S.Q., Zhang M.Y., Zhang H.L., Zheng B., He K.B. Model vs. observation discrepancy in aerosol characteristics during a half-year long campaign in Northeast China: the role of biomass burning. *Environmental Pollution*, 2021, 269: 116167.

---

## Editor Decision (ED1)

**Final decision-acp-2023-51**

Based on the detailed comments of the two experts in the field, and after my consideration, the manuscript is of satisfactory atmospheric interest to merit publication as a Measurement report in *Atmospheric Chemistry and Physics.*

The authors have correctly addressed all the questions/comments raised by the reviewers, and me, and modified the manuscript according to the suggestions. Besides, the revised manuscript has been checked again by the Anonymous referee 2, who also supports the publication.

The manuscript can be accepted for publication in ACP.

**Additional comments-acp-2023-51**:
Please, correct the references according to instructions:
https://www.atmospheric-chemistry-and-physics.net/submission.html#references

Sections 4 & 5 should be merged into one as: *"Conclusions and atmospheric implications"*. This is also a suggestion of the referee:
*"I noticed that a new section entitled "Implications" was incorporated into the revised manuscript. However, it seems odd to me that this section was placed after the "Conclusions" section. Based on the content of the section "Implications", I would suggest to authors to incorporate the new content within the Conclusions and rename the last section "Conclusions and Outlook" or similar«*

Sincerely,

Irena Grgić